# Patient-Reported Measures Associated with the Return to Pre-Injury Levels of Sport 2 Years after Anterior Cruciate Ligament Reconstruction

**DOI:** 10.3390/jfmk8010028

**Published:** 2023-02-23

**Authors:** Zakariya H. Nawasreh, Mohammad A. Yabroudi, Anan B. Al-Shdifat, Sakher M. Obaidat, Sharf M. Daradkeh, Mohamed N. Kassas, Khaldoon M. Bashaireh

**Affiliations:** 1Division of Physical Therapy, Department of Rehabilitation Sciences, Jordan University of Science and Technology (JUST), Irbid 22110, Jordan; 2Department of Physical Therapy and Occupational Therapy, Faculty of Applied Medical Sciences, The Hashemite University, Zarqa 13133, Jordan; 3Department of Special Surgery, College of Medicine, Jordan University of Science and Technology (JUST), Irbid 22110, Jordan

**Keywords:** anterior cruciate ligament, athletes, knee, return to sports, patient-reported outcomes

## Abstract

The International Knee Documentation Committee Subjective Knee Form (IKDC2000) and the Knee Injury and Osteoarthritis Outcome Score (KOOS) are knee-specific measures. However, their association with a return to sports after anterior cruciate ligament reconstruction (ACLR) is unknown. This study aimed to investigate the association between the IKDC2000 and the KOOS subscales and the return to the same pre-injury level of sport two years after ACLR. Forty athletes that were two years post-ACLR participated in this study. Athletes provided demographic information, filled out the IKDC2000 and KOOS subscales, and indicated whether they returned to any sport and whether they returned to the same pre-injury level (same duration, intensity, and frequency). In this study, 29 (72.5%) athletes returned to play any sport and eight (20%) returned to the same pre-injury level. The IKDC2000 (r: 0.306, *p* = 0.041) and KOOS quality of life (KOOS-QOL) (r: 0.294, *p* = 0.046) significantly correlated with the return to any sport, but it was age (r: −0.364, *p* = 0.021), BMI (r: −0.342, *p* = 0.031), IKDC2000 (r: 0.447, *p* = 0.002), KOOS-pain (r: 0.317, *p* = 0.046), KOOS sport and recreation function (KOOS-sport/rec)(r: 0.371, *p* = 0.018), and KOOS QOL (r: 0.580, *p* > 0.001) that significantly correlated with a return to the same pre-injury level. High KOOS-QOL and IKDC2000 scores were associated with returning to any sport, and high KOOS-pain, KOOS-sport/rec, KOOS-QOL, and IKDC2000 scores were all associated with returning to the same pre-injury level of sport.

## 1. Introduction

Anterior cruciate ligament (ACL) rupture is the most commonly injured knee ligament among young athletes participating in multidirectional sports [1]. Rupturing of the ACL has negative long-term consequences, as individuals with an ACL rupture exhibit dynamic knee instability, strength deficits, decreased knee functional performance, and adapt to the injury with abnormal neuromuscular strategies and an aberrant gait pattern, limiting their ability to participate in daily living and sports [2]. Currently, ACL reconstructive surgery (ACLR) is the gold-standard intervention for athletes who have the desire to return to multidirectional sports activities [3]. ACLR has been advocated to help restore mechanical knee stability [4] as well as improve knee function [5], both of which are crucial for returning to multidirectional sports and preserving the integrity of the knee joint structures. Returning to pre-injury levels of sport without sustaining a second ACL injury has been advocated as a successful outcome following the ACLR [6]. The reported rate of return to the same pre-injury sports, however, is highly variable and less than optimal [7]. A systemic review and meta-analysis study reported that 65% of athletes returned to their pre-injury level of sports following ACLR, whereas only 55% of athletes returned to their previous competitive level [7].

Furthermore, Webster et al. [8] found that only 24% of patients who expected to return to sports before the reconstructive surgery (84%) returned to their pre-injury sport level 12 months after ACLR surgery. Therefore, identifying factors associated with a return to the pre-injury level of sport may help clinicians develop tailored rehabilitation programs that could potentially increase the likelihood of their patients returning to their sports following ACLR.

Previous studies have identified multiple factors related to athletes’ demographics (i.e., sex, body mass index (BMI), smoking status) and functional performances (i.e., high hop limb symmetry indexes) that are associated with a return to the pre-injury level of sport [7,9,10,11,12]. Patient-reported outcome measures have been widely used in ACL research to assess patients’ perception about their health status after ACLR surgery. The most commonly used measures are the International Knee Documentation Committee Subjective Knee (IKDC2000) and the Knee Injury and Osteoarthritis Outcome Score (KOOS) [13,14]. Both the IKDC2000 [13] and the KOOS [13,14] are validated knee-specific patient-reported measures in patients with ACL rupture and ACLR [15]. The IKDC2000 was developed to assess changes in patient symptoms, complaints, and knee functions due to the knee injury, and the problems during daily living and sporting activities [15]. The KOOS was developed to assess the short- and long-term effects of the knee injury, as well as the knee-related quality of life and function in sport and recreation [16]. However, the relationship between patients’ IKDC2000 and KOOS subscales scores and their return to sport following ACLR has not been thoroughly investigated. Therefore, studies that investigate the relationship between patient-reported measures and the return to the pre-injury level of sport may aid clinicians in determining the contributions of these variables to the return to sport following ACLR. Furthermore, it can assist clinicians in identifying athletes’ impairments and functional limitations, thus addressing these factors during the post-operative ACL rehabilitation program, hence increasing the rate of a patient’s return to sport.

The purpose of this study was to investigate whether the IKDC2000 and KOOS subscales correlate with athletes returning to any sport and also whether they return to the same pre-injury level two years following ACLR. It was hypothesized that scoring high on the IKDC2000 and IKDC subscales correlates with the return to any sport and the return to pre-injury levels of sport following ACLR.

## 2. Materials and Methods

### 2.1. Participants

Medical records of patients who had ACL injuries within 3 years were screened, and patients were contacted to participate in this study. In total, 40 male athletes aged between 18 and 55 years old with an average of 2 years after primary unilateral ACLR participated in this study between November 2020 and August 2021 (Figure 1). Athletes who regularly participated in Level I or II sport, per the IKDC2000 guidelines, for more than 50 h per year before their ACL rupture were eligible to participate in this study [17]. Level I sport includes jumping, cutting, and pivoting maneuver activities (i.e., soccer, basketball), whereas Level II activities include lateral movement maneuvers (i.e., tennis). Athletes who had unilateral ACLR surgery with the involvement of meniscus injury were also eligible to participate. Athletes who had a revisional ACLR surgery, concomitant knee ligamentous injury, osteochondral lesion, knee osteoarthritis, or serious lower extremity injuries, including fractures and deformities, were excluded from participation in the study. Athletes were recruited from several clinics throughout the country to maximize the generalizability of the study’s findings. The protocol of the current study was approved by the institutional review board of Jordan University of Science and Technology and King Abdullah University Hospital. All athletes provided informed consent before study participation.

### 2.2. Data Collection

Athletes provided demographic information (age, weight, and height), an injury profile (date of injury, concomitant injuries), a pre-injury activity profile (type, level, and participation duration and frequency), and surgical information (date of surgery, graft type, meniscus surgery). A certified physical therapist assessed whether athletes were eligible to participate in this study. Athletes also filled out the Arabic version of the International Knee Documentation Committee Subjective Knee Form (IKDC2000) and the Knee Injury and Osteoarthritis Outcome Score (KOOS) [18]. The IKDC2000 is a valid and reliable knee-specific subjective outcome measure for Arabic patients with ACLR that evaluates the overall knee function [19]. The IKDC2000 consists of three domains: symptoms, sporting activity, and knee function [20]. The symptoms subscale is useful for assessing problems including knee discomfort, weakness, swelling, and giving away. The sporting activity subscale concentrates on activities such as stair climbing, moving up from a seat, squatting, and jumping, whereas the knee function subscale focuses on basic questions related to the state of the knee function before and after the injury. The total score is calculated by adding the scores of the three subscales and then converting them into a 0–100 scale, with a lower score indicating lower knee function [21]. KOOS is a valid and reliable outcome measurement that assesses patients’ perception of their knee and related problems after ACL injury and reconstructive surgery [18]. It consists of five subscales (KOOS-pain: 9 items; KOOS-symptoms: 7 items; KOOS activity of daily living function (KOOS-ADLS): 17 items; KOOS sport and recreation function (KOOS-sport/rec): 5 items; and quality of life (KOOS-QOL): 4 items). Each item on each subscale has five possible response options, with scores ranging from 0 (no problems) to 4 (extreme problems) based on a Likert scale, and the score for each subscale is calculated as the sum of all the items included and then converted to a 0–100 scale, with 0 indicating severe knee problems and 100 indicating no knee problems and no pain.

### 2.3. Return to Sports

Athletes indicated whether they have returned to participate in any sports after reconstructive surgery by answering, “Did you return to sports? (Yes or No)?” Athletes who answered yes to the preceding question were then asked if they have returned to participate at the same pre-injury level, including the same frequency, duration, and intensity: “Did you return to your pre-injury sport levels, this means the same frequency, duration, and intensity of sports? (Yes or No)?”

### 2.4. Statistical Analysis

An independent *t*-test was used to determine if there were significant differences in continuous measures (age, BMI, time from injury to surgery) between those who did (Returned) and did not (Not-Return) return to participate in the same pre-injury sporting activities. The Chi-square test was used to determine if there were significant differences between groups for the dichotomized measures. Pearson correlation coefficients were calculated to assess the relationship between athletes’ demographic (age and BMI), IKDC2000, and KOOS subscales measures and the return to any sport, as well as the return to the same pre-injury sport levels (same duration, intensity, and frequency) following ACLR. Pearson coefficient (r) denotes the strength and direction of an association between variables. It ranges between −1 and +1, with (r < 0.1) indicating no correlation, (0.1 > r > 0.3) indicating a weak correlation, (0.3 > r > 0.5) indicating a moderate correlation, and r > 0.5 indicating a strong correlation [22,23]. SPSS (Version 25.0, IBM Company, Chicago, IL, USA) was used to analyze the data, with the significance level set at a *p*-value of <0.05.

## 3. Results

### 3.1. Demographic Characteristic

Forty male athletes participated in this study at a follow-up time of 2.13 ± 0.44 years after ACLR. All athletes had single-bundle hamstring tendon autograft reconstruction surgery using the medial portal technique. Eighteen athletes had meniscus injuries, of which fourteen athletes had partial meniscectomy at the time of ACLR surgery. In total, 29 (72.5%) athletes returned to play any sport 2 years after ACLR, but only 8 (20%) returned to their pre-injury level. Table 1 illustrates athletes’ demographics, time from injury to surgery, time from surgery to study participation, and the number of athletes with a meniscus injury. There were significant differences between the groups regarding the athletes’ BMI (*p* = 0.026) and the number of meniscus injuries (*p* = 0.022) (Table 1).

### 3.2. Return to Sports

The athletes’ BMI (r = −0.366, *p* = 0.034; Figure 2A) and the IKDC2000 score (r = 0.306, *p* = 0.041, Figure 2C) had a moderate correlation, whereas the KOOS-QOL score (r = 0.294, *p* = 0.046; Figure 2B) had a weak correlation with the return to any sport 2 years after ACLR (Table 2). All of the athletes’ ages (r = −0.364, *p* = 0.021, Figure 3A), BMI (r = −0.342, *p* = 0.03, Figure 3B), KOOS-pain scores (r = 0.317, *p* = 0.046, Figure 3C), the KOOS-sport/rec scores (r = 0.371, *p* = 0.018, Figure 3D), and the IKDC2000 scores (r = 0.447, *p* = 0.002, Figure 3F) had a significant moderate correlation, and the KOOS-QOL score (r = 0.580, *p* < 0.001, Figure 3E) had a significant strong correlation with the return to participate in the same pre-injury level of sport two years after ACLR (Table 2).

## 4. Discussion

The rate of return to any sport was 72.5%, while only 20% of athletes returned to participate at their pre-injury level regarding the frequency, duration, and intensity at 2.13 ± 0.44 of the follow-up time after ACLR. The return to any sport 2 years after ACLR was moderately negatively associated with the BMI, moderately positively associated with a high IKDC2000 score, and weakly positively associated with a high KOOS-QOL score. However, the return to a pre-injury level was moderately negatively associated with the age of the athlete and their BMI, moderately positively associated with a high KOOS-pain score, KOOS sport/rec, and a high IKDC2000 score; and strongly positively associated with a high KOOS-QOL score 2 years after ACLR. The findings of this study contribute to the identification of modifiable variables by rehabilitation, which can be moderately to strongly associated with the return to the same pre-injury level of sport 2 years after ACLR. The findings of this study may also help clinicians to incorporate treatment strategies to resolve knee impairments and enhance athletes’ functional performance during the post-operative ACL rehabilitation program, which will increase the athletes’ ability to return to the same pre-injury level following ACLR.

In this study, the rate of return to any sports was high; however, the rate of return to the same pre-injury level was lower than that reported in previous studies (65%) [7,12,24]. The latter might be attributed to the use of a strict definition of “return to pre-injury levels of sport”, which accounts for the return to participation at the same frequency, intensity, and duration as before the ACL injury. Returning to sports after ACLR has been inconsistently defined in the literature using different methods, including the return to participation at the same pre-injury level using a single global question, “Did you return to participate in the same pre-injury sport levels?” [25,26], or by using patient-reported outcome measures such as the Tegner scale [27] or the Marx Activity Rating Scale [28] and whether patients returned to the same number of hours of participation per year as before their injury [29]. Another factor that might have contributed to the lower rate of return is the study’s sample, which consisted primarily of older athletes (sample mean age: 26.3). Older athletes usually tend to have less motivation to return to sports due to increased work or family commitments, as well as changes in lifestyle or socioeconomic status [30]. Other factors related to the status of the meniscus may potentially have contributed to the lower rate of the return to sports. In the current study, more athletes with meniscus injuries did not return to any sports, and meniscal injury has previously been linked to this after ACL injury and ACLR [31,32].

Our study revealed that athletes’ BMI had a moderate negative correlation with both a return to any sport and the return to participation at the same pre-injury level 2 years following ACLR. Athletes with a lower BMI returned to any sport and also returned to participate at the same pre-injury level. This finding is consistent with previous studies, which found that patients with a BMI between 20 and 25 kg/m^2^ were almost two times more likely to return to pre-injury sporting participation compared to patients with a BMI of more than 26 kg/m^2^ [33]. Another study by Dunn and colleagues reported that a lower BMI was associated with a return to a higher activity level 2 years after ACLR [34]. However, the findings of our study contradict the finding of another study by Mardani et al. [35], which found that the rate of the return to sports was not significantly different across groups of various BMI categories. In Mardani’s study, however, the majority of patients had a normal BMI of 21–25 kg/m^2^ [35]. It has been shown that individuals with a high BMI demonstrated a slower extensor muscle strength recovery and a decreased quadriceps muscle symmetry index after ACLR [36], and a progressive deterioration in their physical functions [37,38].

The age of an athlete also had a moderate negative association with a return to the same pre-injury level of sport. However, the age of athletes was not associated with returning to any sport after ACLR. This may indicate that older athletes are less likely to return to participating at the same pre-injury level 2 years after ACLR, and this finding is consistent with previous studies [33,39]. It is possible that older athletes did not return to participate at the same level as before their injury due to a change in lifestyle, fewer opportunities to participate in activities, socioeconomic status, and psychological factors, such as fear of re-injury if they returned to play in a highly competitive manner [33,40]. It is worth noting that older athletes who have had ACLR may opt to deliberately reduce their level and only participate in a sport in order to maintain a healthy lifestyle. This might explain why age was not a discriminator for returning to any sport following ACLR, as returning to any sport may include returning to light activity, such as running, that does not require as much physical ability.

In this study, the KOOS-QOL score had a positive weak association with returning to any sport and a positive strong association with a return to the same pre-injury level two years after ACLR, whereas the IKDC2000 score had a positive moderate association with both returning to any sport and returning to participation at the same level. This indicates that athletes with high awareness of their knee problems, knee difficulties, and less confidence in their knees following ACLR are less likely to return to any sport or to the same pre-injury sport levels. A study by Webster and colleagues found that athletes who met the threshold value for IKDC2000 were more likely to return to competitive sport 12 months after ACLR [41]. Another study by Webster and Feller [42] found that athletes with high IKDC2000 scores (≥95) had three times the odds of returning to their sport following ACLR. According to Logerstedt et al. [43], athletes who scored low on the IKDC2000 did not pass the return to sport criteria after ACLR, whereas athletes who scored within normal IKDC2000 ranges demonstrated high quadriceps and single-legged hop symmetry indexes and scored highly on patient-reported measures compared to those who scored below normal IKDC200 scoring ranges. Returning to participation in any sport or at the same pre-injury level was associated with scoring high on the KOOS-QOL subscale. This may indicate that athletes who scored high on the KOOS quality of life did not experience knee problems, did not modify their lifestyle, and demonstrated no lack of knee confidence or difficulty as a result of the ACLR surgery. This, in turn, may have helped them to return to their sports. It was found that patients with better KOOS-QOL scores were more likely to return to their sports following ACLR [44]. Anand et al. [45] found that patients who returned to the same level of activity after revisional ACLR scored significantly higher on both the KOOS-QOL and IKDC2000 measures compared with those who did not return. The KOOS-QOL and IKDC2000 are both patient-perceived measures that assess different constructs of knee symptoms and function that are relevant to the return to sport after ACLR, which may explain the study’s findings [46]. The notable finding of this study is that the Pearson coefficients for the KOOS-QOL and IKDC2000 measures were higher for patients returning to participate at the same pre-injury level of their sport compared to just returning to any sport. This may indicate that athletes with knee problems, functional limitations, and a lack of confidence in their knee as a result of reconstructive surgery may be unable to participate in high-demand and strenuous sports at the same frequency, intensity, and duration as before the ACL rupture. Furthermore, there might be a group of patients who have modified the way they approach their sports as a result of their knee difficulties and functional status in order to prevent further knee injuries.

In this study, the KOOS pain and KOOS sports/recreation scores had a positive moderate association with returning to participation at the same pre-injury level of sport. In this study, too, athletes who returned to the same pre-injury level may have demonstrated no knee pain or difficulty while performing high-demand tasks. Knee pain has frequently been reported as a cause for not returning to pre-injury activity levels following ACLR [47]. In a systemic report and meta-analysis study, Andrade and colleagues found that patients who returned to sports following ACLR scored higher on the IKDC2000, the KOOS for quality of life, and the KOOS sports/recreation subscales than those who did not return [48]. Furthermore, a previous study reported that the KOOS pain score had a negative association with the return to sports after the injury scale (ACL-RSI) score in patients who returned to sports 7 months after ACLR [49]. This suggests that patients with low pain were psychologically ready to return to sports and did not experience knee pain while participating. The associations between the KOOS pain and the sports/recreation subscales, as well as the return to participation in the same pre-injury level of sport in this study may be due to the fact that these measures examined knee symptoms and functional deficits while performing high-demand tasks (i.e., squatting, running, twisting, and pivoting). The KOOS subscales are patient-reported measures that are valid, reliable, and responsive in both short-term and long-term follow-ups for a variety of knee injuries. Except for the KOOS-symptoms and KOOS-ADLS subscales, all of the KOOS subscales were associated with a return to the same pre-injury level in the current study. This might be because patients tend to have no knee symptoms, and so they can perform their daily living activities without difficulties by the time of the two-year follow-up after ACLR.

The findings of this study could have clinical implications as patient-reported outcomes, including the IKDC2000 and KOOS subscales, could be used as clinical measures for identifying athletes who elect to return to pre-injury sport levels at the same frequency, intensity, and duration following ACLR. The KOOS subscales and IKDC2000 are clinical measures that require a brief time to be filled. Therefore, clinicians may utilize them as a discriminator in the rehabilitation setting to identify patients with ACLR who can return to sports. Furthermore, the findings of this study can help clinicians in incorporating individualized therapeutic training into the post-operative rehabilitation program to resolve knee pain and difficulties. Additionally, clinicians may address knee functional impairments, especially while performing high-demand tasks that replicate sporting activities, to increase the likelihood of returning to participation in pre-injury sporting activities following ACLR. Furthermore, patients with high BMI might be recommended to reduce their body weight to heighten the possibility of returning to participate in their pre-injury sport levels after ACLR.

It is worth noting that neither group demonstrated differences in either the KOOS subscales or the IKDC2000 measures. The current study sample consisted only of male athletes who participated in recreational sporting activities, which may explain why the KOOS and IKDC scores were not different across groups.

This study has limitations, including the lack of female athlete participants. As a result, the study’s findings cannot be generalized to include female athletes with ACLR. Furthermore, this is a cross-sectional study, and there is a lack of longitudinal data to better identify factors that are associated with the return to participation in the same pre-injury levels of sport following ACLR. Another limitation of the current study is the lack of control over the post-operative ACL rehabilitation programs (duration and quality), as the participants were recruited from different rehabilitation clinics across the country. Furthermore, the reasons for not returning to sport were not controlled in this study.

## 5. Conclusions

In this study, the rate of the return to the same pre-injury level of sport following ACLR was low compared to that reported in the literature. The current study helped identify patients’ demographics and patient-reported measures that are associated with the return to any sports, plus the return to participation at the same level 2 years after ACLR. Athletes’ BMI and KOOS-QOL and IKDC2000 scores were associated with a return to any type of sport 2 years after ACLR, whilst athletes’ ages, BMI, KOOS-pain score, KOOS-sport/rec score, KOOS QOL, and IKDC2000 measures were associated with returning to the same pre-injury level 2 years after ACLR. The findings of this study may help clinicians to be aware of the athletes’ demographic and the clinical patient-reported measures that have associations with the return to pre-injury levels. Clinicians may also need to implement proper rehabilitation programs to resolve athletes’ knee impairments, improve their awareness of knee function, and advise athletes to monitor their body weight in order to increase the likelihood of returning to their pre-injury level of sport following ACLR.

## Figures and Tables

**Figure 1 jfmk-08-00028-f001:**
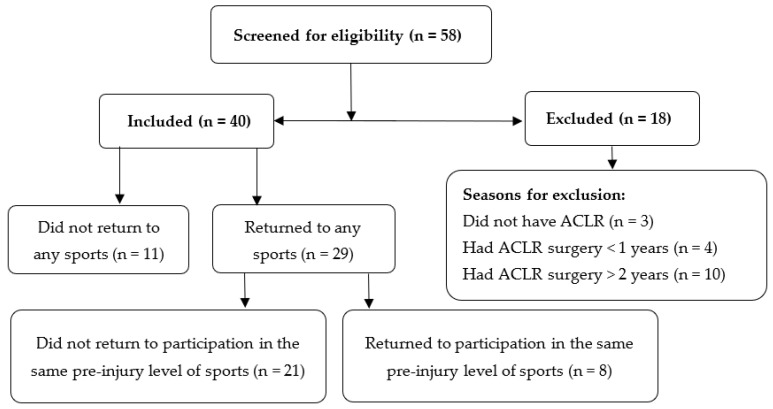
Flow diagram for athletes’ participation in this study.

**Figure 2 jfmk-08-00028-f002:**
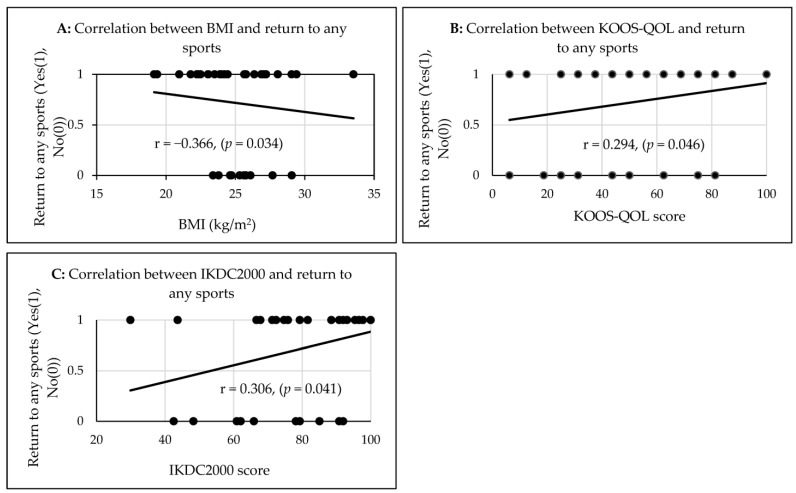
The correlations between the athletes’ BMI (**A**), KOOS-QOL (**B**), and IKDC2000 (**C**) with the return to any sports two years after ACLR.

**Figure 3 jfmk-08-00028-f003:**
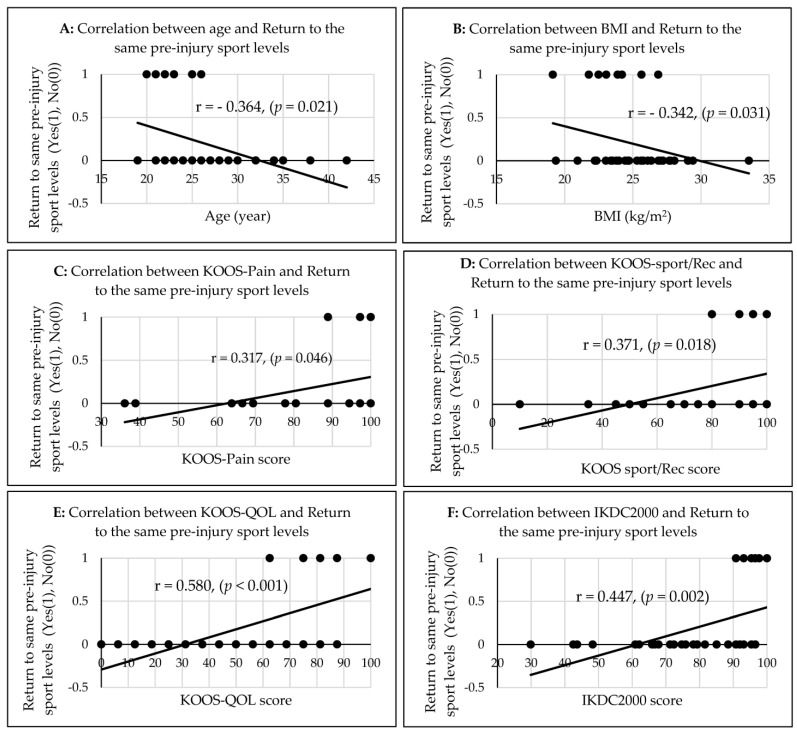
The correlations between the athletes’ age (**A**), BMI (**B**), KOOS-Pain (**C**), KOOS-Sport/rec (**D**), KOOS-QOL (**E**), and IKDC2000 (**F**) with the return to the same pre-injury sport levels two years after ACLR.

**Table 1 jfmk-08-00028-t001:** Demographic characteristics, KOOS subscales, and IKDC2000 of athletes who did and did not return to any sports groups 2 years after ACLR (Means (SD)).

	Did Not Return to Any Sports (*n* = 11)	Returned to Any Sport (*n* = 29)	*p*-Value (95%CI)
Age	28.36 (5.33)	25.54 (4.42)	0.113 (−6.35–0.70)
BMI	26.71 (2.97)	24.54 (2.47)	0.026 (−4.05–−0.30) *
Time from injury to surgery (month)	23.50 (6.26)	13.19 (2.70)	0.084 (−22.10–1.47)
Time from surgery to study participation (month)	26.64 (4.63)	25.89 (5.65)	0.701 (−4.63–3.14)
Meniscus injury (Yes/No)	(6/5)	(11/17)	0.022
KOOS-pain score	85.86 (13.52)	88.29 (17.38)	0.679 (−14.28–9.41)
KOOS-symptoms	57.47 (14.76)	64.67 (14.13)	0.166 (−17.51–3.11)
KOOS-ADLs	91.44 (10.82)	92.33 (14.66)	0.857 (−10.79–9.01)
KOOS-Sport/Rec score	75.00 (21.45)	82.14 (22.09)	0.366 (−22.94–8.66)
KOOS-QOL score	41.47 (27.43)	57.81 (24.63)	0.079 (−34.65–1.99)
IKDC2000 score	71.20 (16.53)	83.21 (16.78)	0.051 (−24.40–0.37)

BMI: Body mass index; KOOS: Knee Injury and Osteoarthritis Outcome Score; ADLs: Activities of daily living IKDC2000: International Knee Documentation Committee; 95%CI: 95% confidence interval; *: indicates significant difference between groups.

**Table 2 jfmk-08-00028-t002:** Pearson correlations between return to sport and return to pre-injury sport levels, and KOOS subscales and IKDC score 2 years after ACLR (Pearson correlations (r (*p*-value))).

	Return to Any Sport	Return to the Same Pre-Injury Sport Levels
Age	(r = −0.221 (*p* = 0.171))	(r = −0.364 (*p* = 0.021)) *
BMI	(r = −0.366 (*p* = 0.034)) *	(r = −0.342 (*p* = 0.031)) *
KOOS-Pain score	(r = 0.077 (*p* = 0.638))	(r = 0.317 (*p* = 0.046)) *
KOOS-Symptoms	(r = 0.178 (*p* = 0.273))	(r = 0.217 (*p* = 0.178))
KOOS ADLs	(r = 0.025 (*p* = 0.879))	(r = 0.268 (*p* = 0.094))
KOOS-Sport/Rec score	(r = 0.132 (*p* = 0.416))	(r = 0.371 (*p* = 0.018)) *
KOOS-QOL score	(r = 0.294 (*p* = 0.046)) *	(r = 0.580 (*p* < 0.001)) *
IKDC2000 score	(r = 0.306 (*p* = 0.041)) *	(r = 0.447 (*p* = 0.002)) *

BMI: Body mass index; KOOS: Knee Injury and Osteoarthritis Outcome Score; ADLs: Activities of daily living IKDC2000: International Knee Documentation Committee; *: indicates significant correlation.

## Data Availability

All the relevant data of this study are available in the Department of Rehabilitation Sciences of Jordan University of Science and Technology, Irbid, Jordan. Data will be available upon a request sent to the corresponding author of this study.

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
