# Peer review of "Patient-Reported Measures Associated with the Return to Pre-Injury Levels of Sport 2 Years after Anterior Cruciate Ligament Reconstruction"

_jfmk, 2023, doi:10.3390/jfmk8010028_

Round 1
Reviewer 1 Report
This is a very interesting study regarding return to play post ACLR . The authors have done a valiant effort. However, further work needs to be done. The authors talk about pre-injury level of activity and they state that this,along demographics, surgery details etc, was documented by a questionnaire and assessed by a certified physiotherapist. Which type of questionnaire? Then the athletes were asked to answer with a yes/no if they returned to same level.Why was the TEGNER score not used to accurately record pre and post injury values? Time from injury to surgery was quite long. Any reason for that? This could be a reason explaining the poor RTP results,even though the given p value is not indicative of that.
Lines 167-169 : The authors make a suggestion about rehabilitaion programmes but this has not been previously discussed in the paper. It is somewhat confusing and usastainable to draw results like that. This sould be rephrased.
Lines 175-177: Use Tegner score
Lines 184-188:Same point with lines 204-210. Please avoid repetition. Moderate negative association is confusing! Is it statistical significant or not?
Lines:219-220: Mild and moderate correlations? Please make a paragraph in your methods to explain what this means otherwise delete. Statistical significane,or not, is a valid count.
Lines 235-237: How was this statement drawn? Please avoid starting sentences with hypothetical quotes
Lines 239-250: This paper concerns primary ACLR . Using references for ACL revision outcomes is confusing.
Limitations are written correctly.
Conclusions section is too extended. Please summarize
Literature is too extensive -for this paper 35 references would be ok.
In summary this is a good paper. In my opinion the authors should have used the Tegner score. Further more the statistical methods should be further explained. Mild ,moderate association should be depicted with p values.
The discussion should be starting with the main finding of this study. The authors should avoid starting sentences with hypothetical quote such as "this might be" etc. Conclusion section should be reduced.
Author Response
We thank the Journal of Functional Morphology and Kinesiology for considering this manuscript “Manuscript ID: jfmk-2147743“ for revision. We also thank both the editor(s) committee and reviewers for their time, efforts, and feedback they provided reviewing this manuscript. We found that the comments and feedback were valuable, helpful, and improved the quality of the manuscript. All the comment were addressed, and changes were made in the manuscript file.
Please be aware that there are some changes to the manuscript as a response to reviewers’ comments and feedback.
The reviewers’ comments were addressed, and the authors responses are below:
Comments and Suggestions for Authors
We would like to thank you for your positive feedback and comments.
This is a very interesting study regarding return to play post ACLR . The authors have done a valiant effort. However, further work needs to be done. The authors talk about pre-injury level of activity and they state that this, along demographics, surgery details etc, was documented by a questionnaire and assessed by a certified physiotherapist. Which type of questionnaire?
Thank you for your comments. This sentence has been corrected as the following: “Athletes answered questions that asked about their demographic information (age, weight, and height), an injury profile (date of injury, concomitant injuries), a pre-injury activity profile (type, level, and participation duration and frequency), and surgery details (date of surgery, graft type, meniscus surgery).”
Then the athletes were asked to answer with a yes/no if they returned to same level. Why was the TEGNER score not used to accurately record pre and post injury values?
Thank you for your comment. In this study, In this study, we included athletes who participated in level I and II sport activities per IKDC classification to control for the sport activities and to exclude those patients who do not play sports. Tegner score was not used as we did not mean to determine the change (increase or decrease) in activity level or intensity. We used in this study a self-reported question about return tp sports that have been used by several previous studies (see the list below):(1–4)
- Nawasreh Z, Logerstedt D, Cummer K, Axe M, Risberg MAMA, Snyder-Mackler L. Functional performance 6 months after ACL reconstruction can predict return to participation in the same preinjury activity level 12 and 24 months after surgery. Br J Sports Med [Internet]. 2018 Mar 27;52(6):375. Available from: http://bjsm.bmj.com/lookup/doi/10.1136/bjsports-2016-097095
- Lentz T a., Zeppieri G, Tillman SM, Indelicato P a, Moser MW, George SZ, et al. Return to preinjury sports participation following anterior cruciate ligament reconstruction: contributions of demographic, knee impairment, and self-report measures. J Orthop Sports Phys Ther [Internet]. 2012 Jan [cited 2014 Jan 28];42(11):893–901. Available from: http://www.pubmedcentral.nih.gov/articlerender.fcgi?artid=3680881&tool=pmcentrez&rendertype=abstract
- Yabroudi MAMA, Bashaireh K, Nawasreh ZHZH, Snyder-Mackler L, Logerstedt D, Maayah M. Rehabilitation duration and time of starting sport-related activities associated with return to the previous level of sports after anterior cruciate ligament reconstruction. Phys Ther Sport Off J Assoc Chart Physiother Sport Med. 2021 May;49:164–70.
- Muller B, Yabroudi MA, Lynch A, Popchak AJ, Lai C-L, van Dijk CN, et al. Return to preinjury sports after anterior cruciate ligament reconstruction is predicted by five independent factors. Knee Surg Sports Traumatol Arthrosc. 2022 Jan;30(1):84–92.
Time from injury to surgery was quite long. Any reason for that? This could be a reason explaining the poor RTP results, even though the given p value is not indicative of that.
Unfortunately, we did not collect these data for this study as we were unaware of this factor prior to the data collection. The time from injury to surgery was long could have resulted form the long time that the patients had to wait to receive the surgery. It could be that patient may have also patient may have decided initially to be managed non-operatively and then switched to operative management (ACLR). We agree with you that long time duration form injury to surgery may have impacted the RTP results in this study as having a long-time duration may have caused degenerative changes or injuries to the joints structures. This, in turn, could affect the RTS results.
Lines 167-169 : The authors make a suggestion about rehabilitaion programmes but this has not been previously discussed in the paper. It is somewhat confusing and usastainable to draw results like that.
This to highlight the clinical implications of the paper findings, which may help to modify and improve our physical therapy practice that take place during rehabilitation program. This has been changed as the following:
“The findings of this study suggest that clinicians may incorporate treatment strategies that help resolve knee impairments and improve athletes’ functional performances after ACLR surgery to increase the athletes’ ability to return to participating in the same pre-injury sport levels post-ACL reconstructive surgery.”
Lines 175-177: Use Tegner score
Activity level classification used in this paper has been previously used to classify the activity level after ACL rupture and ACLR. These are some papers that used the classification:
[1] E.H. Hartigan, A.D. Lynch, D.S. Logerstedt, T.L. Chmielewski, L. Snyder-Mackler, Kinesiophobia after anterior cruciate ligament rupture and reconstruction: noncopers versus potential copers., J. Orthop. Sports Phys. Ther. 43 (2013) 821–32. https://doi.org/10.2519/jospt.2013.4514.
[2] H. Grindem, D. Logerstedt, I. Eitzen, H. Moksnes, M.J. Axe, L. Snyder-Mackler, L. Engebretsen, M.A. Risberg, Single-legged hop tests as predictors of self-reported knee function in nonoperatively treated individuals with anterior cruciate ligament injury., Am. J. Sports Med. 39 (2011) 2347–54. https://doi.org/10.1177/0363546511417085.
[3] Z. Nawasreh, D. Logerstedt, K. Cummer, M.J. Axe, M.A. Risberg, L. Snyder-Mackler, Do Patients Failing Return-to-Activity Criteria at 6 Months after Anterior Cruciate Ligament Reconstruction Continue Demonstrating Deficits at 2 Years?, Am. J. Sports Med. 45 (2017) 1037–1048. https://doi.org/10.1177/0363546516680619.
[4] H. Kan, Y. Arai, M. Kobayashi, S. Nakagawa, H. Inoue, M. Hino, S. Komaki, K. Ikoma, K. Ueshima, H. Fujiwara, I. Yokota, T. Kubo, Fixed-flexion view X-ray of the knee superior in detection and follow-up of knee osteoarthritis., Medicine (Baltimore). 96 (2017) e9126. https://doi.org/10.1097/MD.0000000000009126.
[5] V. Wright, Biomechanics of the Knee. With Application to the Pathogenesis and the Surgical Treatment of Osteoarthritis., Ann. Rheum. Dis. 37 (1978) 105–106.
[6] T.D. Rosenberg, L.E. Paulos, R.D. Parker, D.B. Coward, S.M. Scott, The forty-five-degree posteroanterior flexion weight-bearing radiograph of the knee., J. Bone Joint Surg. Am. 70 (1988) 1479–1483.
[7] R. Riddick, Energetics of Human Running: A Mechanistic Perspective, 2017.
Lines 184-188: Same point with lines 204-210. Please avoid repetition. Moderate negative association is confusing! Is it statistical significant or not?
Repetitions have been removed as the following: Line 184-188: Another reason could be related to the study’s sample, which included primarily older athletes (sample mean age: 26.3), Line 204-2010: Athletes’ age associated also with a return to the same pre-injury sport levels.
Moderate negative association was used to indicate the strength and direction of the association, it was removed from the paper to lower confusions.
Lines:219-220: Mild and moderate correlations? Please make a paragraph in your methods to explain what this means otherwise delete. Statistical significane,or not, is a valid count.
We added to the statistical section information to explain the Pearson coefficient strength and direction as the following: “Pearson coefficient (r) denotes the strength and direction of an association between variables. It is varying between –1 and +1, with (r > 0.1) indicates no association, (0.1>r>0.3) as small, (0.3> r >0.5) as medium, and r>0.5 as large association.(29).”
We meant to include the strength of the association (mild, moderate, and strong) as being statically significant is not enough to indicate the strength of the association.
Lines 235-237: How was this statement drawn? Please avoid starting sentences with hypothetical quotes.
This sentence has been modified as the following : This may indicate that athletes who scored highly on KOOS quality of life did not experience knee problems, did not modify their lifestyle, and demonstrated no lack of knee confidence or difficulty due to the ACLR surgery, which in turn helped them to their return to their sports.
Lines 239-250: This paper concerns primary ACLR . Using references for ACL revision outcomes is confusing.
We indicated that findings of Anand et al.[49] as it’s findings supports the findings of our study even after revisional ACLR surgery. The co-authors of the study do not see a problem in keeping this statement. “Another study by Anand et al.[49] reported that patients who had returned to the same sport levels activity after revisional ACLR scored significantly higher on both the KOOS-QOL and IKDC2000 measures compared with those who did not return.” We would also to highlight to the reviewer that anything after statement for Anand et al [49] (line 242-255) is not for revisional ACLR.
Limitations are written correctly.
Thank you for commenting on that.
Conclusions section is too extended. Please summarize
It has been summarized.
Literature is too extensive -for this paper 35 references would be ok.
We lower it to 45 refrences.
In summary this is a good paper. In my opinion the authors should have used the Tegner score. Further more the statistical methods should be further explained. Mild ,moderate association should be depicted with p values.
The discussion should be starting with the main finding of this study. The authors should avoid starting sentences with hypothetical quote such as "this might be" etc. Conclusion section should be reduced.
Thank you for your positive feedback and comments. We tried our best to address your comments. We added to the statistical section detailed to explain the strength of association. We worked on avoiding hypothetical quotes and summarized the conclusions.

Reviewer 2 Report
January 19, 2023
Dear authors,
I recently had the opportunity to review your manuscript "Patient-reported measures associate with return to pre-injury level of sport 2 years after anterior cruciate ligament reconstruction." I commend you on a well-written and easy to follow study. It is presented clearly. I do have a general concern that your study is not novel. These associations have been demonstrated several times previously. Nevertheless, these are good data to have published, for systematic review or meta-analysis purposes. I have detailed a few, more specific, comments below. I hope they are helpful.
Specific comments:
-Why is BMI reported as a predictor in the abstract? This was not in the purpose statement. I would suggest omitting all BMI inferences, as this is beside the point.
-p = .066 is not significant (abstract).
-Lines 45-51—Can summarize these studies, instead of rote regurgitation.
-Lines 52-61—This content is not related to your purpose. Would suggest deleting.
-New paragraph on line 62 to introduce IKDC and KOOS.
-Line 80—Please replace ‘of’ with ‘post.’
-Line 137—Should be r < .10.
-Statistical analysis subsection—what is considering a meaningful effect size? No a priori p? I’m ok if you elect to not use p values a priori, but if you do so, then you should not use ‘statistical significance’ language.
-Results—Could you provide the pearson R between the surveys? Is it possible they are explaining the same thing?
-Lines 148-149—Use exact p values for these comparisons, if you choose to use p values.
-Table 2—Why are some variables bolded and others not? Need to include the cutoff of what is considered meaningful.
-Line 174—65% returned at 2 years, or at any time?
-Line 282—Not ‘who can return to sport.’ It would be ‘who elects to return to sport.’
-Line 292—Please delete the word “all.”
Author Response
We thank the Journal of Functional Morphology and Kinesiology for considering this manuscript “Manuscript ID: jfmk-2147743“ for revision. We also thank both the editor(s) committee and reviewers for their time, efforts, and feedback they provided reviewing this manuscript. We found that the comments and feedback were valuable, helpful, and improved the quality of the manuscript. All the comment were addressed, and changes were made in the manuscript file.
Please be aware that there are some changes to the manuscript as a response to reviewers’ comments and feedback.
The reviewers’ comments were addressed, and the authors responses are below:
Comments and Suggestions for Authors
January 19, 2023
Dear authors,
I recently had the opportunity to review your manuscript "Patient-reported measures associate with return to pre-injury level of sport 2 years after anterior cruciate ligament reconstruction." I commend you on a well-written and easy to follow study. It is presented clearly. I do have a general concern that your study is not novel. These associations have been demonstrated several times previously. Nevertheless, these are good data to have published, for systematic review or meta-analysis purposes. I have detailed a few, more specific, comments below. I hope they are helpful.
Thank you for your positive comments and feedback.
Specific comments:
-Why is BMI reported as a predictor in the abstract? This was not in the purpose statement. I would suggest omitting all BMI inferences, as this is beside the point.
The BMI and age were removed form the abstract.
-p = .066 is not significant (abstract).
The p values for both IKDC2000 (r:0.306, p=0.041), and KOOS quality-of-life (QOL) (r:0.294, p=0.046) were double checked and corrected throughout the paper
-Lines 45-51—Can summarize these studies, instead of rote regurgitation.
This has been changes into:
According to a systemic review and meta-analysis study, 65% of players returned to the previous level of sports, while only 55% of players returned to their previous competitive level after ACLR.(6)
-Lines 52-61—This content is not related to your purpose. Would suggest deleting.
-New paragraph on line 62 to introduce IKDC and KOOS.
We tried to provide data about studies that looked at predictor factors for return to sport after ACLR in this section. However, this section has been deleted.
-Line 80—Please replace ‘of’ with ‘post.’
It has been changed.
-Line 137—Should be r < .10.
It has been changed.
-Statistical analysis subsection—what is considering a meaningful effect size? No a priori p? I’m ok if you elect to not use p values a priori, but if you do so, then you should not use ‘statistical significance’ language.
Thank you for the comment. Our understanding of conducting a statistical test for correlation is that p-value tells us whether the correlation is significantly different from zero (which means no correlation); and the absolute value of the correlation coefficient ( r ) is used to determine the effect size of the association (an effect size that summarizes the strength and direction of the association).
-Results—Could you provide the pearson R between the surveys? Is it possible they are explaining the same thing?
I think you mean the collinearity between return to any sport and return to the same pre-injury level of sports (r=0.298, p=0.058)
-Lines 148-149—Use exact p values for these comparisons, if you choose to use p values.
Exact p value was eased for each measure.
-Table 2—Why are some variables bolded and others not? Need to include the cutoff of what is considered meaningful.
Bold was removed from table 1 and 2.
-Line 174—65% returned at 2 years, or at any time?
following ACLR has been added as the exact time was not indicated in Ardern study.
-Line 282—Not ‘who can return to sport.’ It would be ‘who elects to return to sport.’
It has been changed.
-Line 292—Please delete the word “all.”
It has been deleted.

Reviewer 3 Report
The present article aims to investigate the association between the IKDC2000 and KOOS sub-scales and the return to the same pre-injury sport levels 2 years after ACLR. However, below a series of suggestions are recommended to the authors for the publication of the manuscript.
The authors present a direct introduction and solid, but somewhat brief content. However, all the content is presented equally, structure the introduction in the following well-differentiated parts and in different paragraphs:
- Conceptual description/introduction and importance of the topic.
- Background: discuss previous and current research in the field.
- Identify the current problem and explain the approach taken to solve it, as well as its usefulness.
- Objective/s and hypothesis
The method used is a simple method with little crumb statistics. But it is correct to respond to the objective that is set. Just one thing, where is the significance level set? Because in line 152 it says that there are significant correlations and 0.034 is significant but 0.055 is not. Review this throughout the entire article.
Table 1 gives a schematic and clear vision of the results but the results of Table 2 should be represented in figures and not in tables. This would give much clearer visual information of the results.
On the other hand, in the methodology he talks about small, medium and large correlations but then he only talks about whether they are significant or not. Review this throughout the article.
Carefully review the conclusions, the interpretations should be in the discussion after sharing the results, or in practical applications. The results should be objective after the data obtained from this manuscript.
Author Response
We thank the Journal of Functional Morphology and Kinesiology for considering this manuscript “Manuscript ID: jfmk-2147743“ for revision. We also thank both the editor(s) committee and reviewers for their time, efforts, and feedback they provided reviewing this manuscript. We found that the comments and feedback were valuable, helpful, and improved the quality of the manuscript. All the comment were addressed, and changes were made in the manuscript file.
Please be aware that there are some changes to the manuscript as a response to reviewers’ comments and feedback.
The reviewers’ comments were addressed, and the authors responses are below:
Comments and Suggestions for Authors
The present article aims to investigate the association between the IKDC2000 and KOOS sub-scales and the return to the same pre-injury sport levels 2 years after ACLR. However, below a series of suggestions are recommended to the authors for the publication of the manuscript.
The authors present a direct introduction and solid, but somewhat brief content. However, all the content is presented equally, structure the introduction in the following well-differentiated parts and in different paragraphs:
Each of the suggested structural item has been modified and stated in different paragraphs.
- Conceptual description/introduction and importance of the topic.
It has been modified in 1st paragraph of the introduction.
- Background: discuss previous and current research in the field.
It has been modified in the 2nd paragraph of the introduction. Just for your knowledge, one of the reviewer suggested to delete the following section as it is not related to the topic of the paper:
“Returning to the pre-injury sport levels is multifactorial.(6) Male young athletes’ with a low BMI, symmetrical hop functional performance, elite sports experience, and a positive psychological response favored returning to pre-injury multidirectional sport.(6) (10) Further, longer durations of rehabilitation programs have resulted in better patient-reported knee outcomes, a higher quality of life, and a greater return to sports activities.(7) While female athletes, those who smoked before ACLR surgery, and having a revisional ACLR were associated with a return to a lower sport level.(6) As many athletes do not return to pre-injury sports activities after ACLR surgery, it is important to identify factors that can contribute to the athletes’ ability to resume their sports. Nawasreh et al(10) found that the hop limb symmetry index at 6 months after ACL-R was a predictor for return to the sport at one and two years, while the Knee Outcome Survey–Activity of Daily Living scale and Global Rating Scores were only predictors for return to the sport at 1 year after ACLR.”
- Identify the current problem and explain the approach taken to solve it, as well as its usefulness.
It has been modified at the end of the 3rd paragraph of the introduction as the following
“Therefore, studies that investigate the association between patient-reported measures and the return to pre-injury sport level may help clinicians to determine the contributions of these measures to the return to sport after ACLR. Furthermore, it can help identify athletes’ impairments and functional limitation, which help clinicians to address these impairments and functional deficits during the post operative ACL rehabilitation program to increase the rate of returning to sports.”
- Objective/s and hypothesis
It has been stated in separate paragraph.
The method used is a simple method with little crumb statistics. But it is correct to respond to the objective that is set. Just one thing, where is the significance level set? Because in line 152 it says that there are significant correlations and 0.034 is significant but 0.055 is not. Review this throughout the entire article.
The following sentence was added at the end of the stats section” Data were analyzed using the SPSS (Version 25.0, IBM Company, Chicago, Illinois, USA), with the significance level set at a p-value of <0.05.”
The p-values were double checked and corrected accordingly throughout the article (abstract, results, and tables).
Table 1 gives a schematic and clear vision of the results but the results of Table 2 should be represented in figures and not in tables. This would give much clearer visual information of the results.
On the other hand, in the methodology he talks about small, medium and large correlations but then he only talks about whether they are significant or not. Review this throughout the article.
The small, medium, and large correlation were changed to weak, moderate and strong correlation in the stat subheading and the effect size for each factor was indicated in the discussion to provide the reader about the contribution (effect size) for each pair of correlation .
Carefully review the conclusions, the interpretations should be in the discussion after sharing the results, or in practical applications. The results should be objective after the data obtained from this article.
The conclusion bas been carefully modified.

Reviewer 4 Report
Please see the attached file.

Author Response
We thank the Journal of Functional Morphology and Kinesiology for considering this manuscript “Manuscript ID: jfmk-2147743“ for revision. We also thank both the editor(s) committee and reviewers for their time, efforts, and feedback they provided reviewing this manuscript. We found that the comments and feedback were valuable, helpful, and improved the quality of the manuscript. All the comment were addressed, and changes were made in the manuscript file.
Please be aware that there are some changes to the manuscript as a response to reviewers’ comments and feedback.
The reviewers’ comments were addressed, and the authors responses are below:
Review of the manuscript titled Patient-Reported Measures Associate with Return to Pre-Injury Level of Sport 2 Years after Anterior Cruciate Ligament Reconstruction.
The most significant disadvantage of the present manuscript is that even though the study aims to investigate the association between the two commonly used patient-rated scales and the return to preinjury level of sport, the authors didn’t use any ACL-specific scale related to sport, like Tegner Activity Level Score or ACL-Return to Sport after Injury (ACL-RSI) scale.
Thank you for your comment. The aim of this study was to determine whether patient-reported measures including the IKDC2000 and KOOS measures associate with return to the same pre-injury sport level. We completely agree with the reviewer that Tegner is an important and commonly used scale for determine RTS after ACLR.
In this study, we included athletes who participate in level I and II sport activities per IKDC classification to control for the sport activities and to exclude those patients who do not play sports. TENGER score was not used which could be a limitation for our study. However, Tegner score was not used as we did not mean to determine the change (increase or decrease) in activity level or intensity. We used in this study a self-reported question about return tp sports that have been used by several previous studies (see the list below):(1–4)
- Nawasreh Z, Logerstedt D, Cummer K, Axe M, Risberg MAMA, Snyder-Mackler L. Functional performance 6 months after ACL reconstruction can predict return to participation in the same preinjury activity level 12 and 24 months after surgery. Br J Sports Med [Internet]. 2018 Mar 27;52(6):375. Available from: http://bjsm.bmj.com/lookup/doi/10.1136/bjsports-2016-097095
- Lentz T a., Zeppieri G, Tillman SM, Indelicato P a, Moser MW, George SZ, et al. Return to preinjury sports participation following anterior cruciate ligament reconstruction: contributions of demographic, knee impairment, and self-report measures. J Orthop Sports Phys Ther [Internet]. 2012 Jan [cited 2014 Jan 28];42(11):893–901. Available from: http://www.pubmedcentral.nih.gov/articlerender.fcgi?artid=3680881&tool=pmcentrez&rendertype=abstract
- Yabroudi MAMA, Bashaireh K, Nawasreh ZHZH, Snyder-Mackler L, Logerstedt D, Maayah M. Rehabilitation duration and time of starting sport-related activities associated with return to the previous level of sports after anterior cruciate ligament reconstruction. Phys Ther Sport Off J Assoc Chart Physiother Sport Med. 2021 May;49:164–70.
- Muller B, Yabroudi MA, Lynch A, Popchak AJ, Lai C-L, van Dijk CN, et al. Return to preinjury sports after anterior cruciate ligament reconstruction is predicted by five independent factors. Knee Surg Sports Traumatol Arthrosc. 2022 Jan;30(1):84–92.
The authors built their conclusions only on self-reported yes or no answers. Also, please don’t get me wrong, but the conclusions are highly biased and predictable because of the choice of scales. For example, let’s take a look at the first question in the IKDC2000 Subjective Knee Evaluation form. The question is, “What is the highest level of activity that you can perform without significant knee pain?” The two best options with the highest scores are: “Very strenuous activities like jumping or pivoting as in basketball or soccer” and “Strenuous activities like heavy physical work, skiing or tennis”. So, any person who returned to sport would choose between those two the best answers. Another example: “How does your knee affect your ability to: go upstairs or go downstairs”. It’s so obvious that patients who returned to sports would always choose the best answers. The situation is pretty much the same with KOOS. The questionnaire assesses how the patient feels about their knee and how well they are able to perform their usual activities. We do not expect that patients who have bad feelings about their knees or patients who have problems with ADL would return to sport on the same level as prior to the injury, as long as they didn’t play chess.
We highly appreciate your comment, and we agree with you as we anticipated correlations between KOOS and IKDC scales with return to sport. In this study, we aimed to determine the strength and direction of the correlations. The results of this study revealed that even with using a Yes and No question, only 20% returned to the same preinjury level.
Line 14: Do the authors mean the subjective part of the IKDC2000?
Thank you for this comment. It has been corrected and added to the abstract and methods .
Line 39: Not only young.
“Young” has been removed.
Line 49: Do the authors mean that patients were expected to return to sport or patients expected to return to sport? I believe it is the second option.
It has been corrected. “Patient expected”
Line 58: There was no information regarding postoperative physiotherapy.
Please take a look at the manuscripts of Królikowska et al. indicating the role of supervised versus nonsupervised postoperative physiotherapy.
Participants were recruited from different physical therapy clinics in the private and public sectors. We believe that the rehabilitation program influences the patients outcomes. However, we did not control for the PT program as it was not the focus of the article. It was indicated as a limitation to this study.
Line 73: Correlation is rather a method of measuring an association (relation, connection). There is absolutely no information concerning the crucial ACL RTS issue, namely the return to sport continuum (for example, Meredith et al. 2021).
The return to sport, according to authors, should be confronted with the ACL RTS continuum (return to sport, return to participation, return to performance). Please face your manuscript with the consensus.
The aim of this study was to assess RTS at 2 years after ACLR and to determine the correlation between patient-reported measures with RTS at 2 years after ACLR. While we commence the importance of RTS continuum, the study design was a cross-sectional study that assess RTS at one time point and to assess the correlation of the factors of interest with the RTS at that time point.
Lines 80-81: Is it a representative group? Did you calculate a minimal sample or carry out a power analysis?
This study is part of a powered project. For the corrections analysis and multiple comparisons, we consulted with a statistician, and he indicated that the statistical analysis used in this study computes the correlation between each pair of variables regardless of number of variables included in the analysis.
Line 82: Were they recreational athletes or professional athletes? If you had added TAS, you would know that.
The participants of this study were mainly recreational athletes.
Figure 1: What were the reasons that 11 patients didn’t return to sport?
We collected data on the reasons for not returning sport and we are working on analyzing the data and writing a manuscript.
Figure 1: What about the return to performance?
We collected data about returning to any sport and returning to the same frequency, duration, and intensity of sports). This could imply returning to participation, playing and performance. We thank you again for highlighting this issue, we will consider that in future studies.
Figure 1: Who was in the primary group before the application of inclusion and exclusion criteria (58 patients?)
In this cross-sectional study, medical records of patients with ACL injury within 3 years were screened and patients were contacted for study participation. We screened 58 patients with ACLR. The inclusion criteria to include athletes 2 yrs post ACLR.
Table 1: What’s pretty unexpected is that there were no differences when comparing results between patients who returned to the sport and those who didn’t. Why wasn’t it highlighted, and your conclusions are based only on correlations that are basically obvious?
Thank you for the comment. Subjects who participated in this study were athletes (mainly recreational athletes), which may explain why groups were not different. We tried to focus on presenting and discussing the results that serve the purpose of the study.
Table 1: In such a small group, it would be better if the results were provided not as a standard deviation, but confidence interval.
95%CI was added to table 1
Table 2: Where did the quantitative data from the “yes” or “no” answers about the return to sport come from?
This concern is not clear to the authors. In this study, patients filled the KOOS-subscales and the IKDC forms and the answered if they retuned to sport (YES/NO). Patients were classified as returned and not returned and their corresponding KOOS and IKDC scores was used in the analysis.
Table 2: The correlations that were noticed are very low. I don’t think they have any meaning. The linear Pearson’s correlation coefficient is calculated to measure the strength and direction of any linear relationships between the selected parameters. The magnitudes of all of the bivariate associations are classified as negligible (0.00-0.30), low (0.31–0.50), moderate (0.51–0.70), high (0.71–0.90), and very high (0.91–1.00) [34].
Additionally, the coefficient of determination, the r-squared (r2 ), might be calculated to give a proportion of variance (fluctuation) of one variable that is predictable from the other variable. Namely, r2 represents the percentage of data points that are the closest to the line of best fit. For example, let's look at the KOOS-Pain Score: r=-0.342; p=0.031). Indeed, because p is smaller than 0.05, we can say that there exists a correlation; however, it’s low (between 0.31 and 0.50).
Also, if we calculate fluctuation (0.342 x 0.342 = 0.117; 0.117 x 100%=12%) we can say that only in 11% of studied patients, the result is an effect of the negative correlation between the studied parameters. So basically, it means nothing.
We completely agree with you. In this study we used the correlation instead of using the prediction due to the limitation of the cross-sectional design of this study. We consulted with the statistician, and he suggested using the correlation since the KOOS and IKDC scores were collected at the same time with the questions asking about returning to sport (yes/No). The correlations in this study ranged between weak to strong based on the classification used in this study (reference: . Cohen , J, Cohen J, Cohen , J, Cohen J. Statistical Power Analysis for the Behavioral Sciences. 2nd ed., editor. Statistical Power Analysis for the Behavioral Sciences. Routledge. https://doi.org/10.4324/9780203771587: L. Erlbaum Associates; 1988.)
The R2 for the measures used in this study ranged between 8% -33.64%, which we believe that are reasonable considering the return to sport is multifactorial. The authors would like to highlight that the purpose of this study was to determine whether the patient-reported measure correlate with return to sport after ACLR and to determine the direction and strength of the correlation. We believe that the sample of the study revealed findings that could help to be aware of the effect size for the KOOS and IKDC measures when they used after ACLR.
The manuscript has so many drawbacks that I didn’t read the discussion.

Round 2
Reviewer 1 Report
-
Author Response
We thank the Journal of Functional Morphology and Kinesiology for considering this manuscript “Manuscript ID: jfmk-2147743“ for 2nd revision. We also thank both the editor(s) committee and reviewers for their time, efforts, and feedback they provided reviewing this manuscript. We found that the comments and feedback were valuable, helpful, and improved the quality of the manuscript. All the comment were addressed, and changes were made in the manuscript file and highlighted in GREEN. The manuscript underwent proofing and copyediting by professional English speaker.
Please be aware that there are some changes to the manuscript as a response to reviewers’ comments and feedback.
The reviewers’ comments were addressed, and the authors responses are below:
Open Review
( ) I would not like to sign my review report
(x) I would like to sign my review report
English language and style
( ) English very difficult to understand/incomprehensible
( ) Extensive editing of English language and style required
( ) Moderate English changes required
(x) English language and style are fine/minor spell check required
( ) I don't feel qualified to judge about the English language and style
|
Yes |
Can be improved |
Must be improved |
Not applicable |
|
|
Does the introduction provide sufficient background and include all relevant references? |
(x) |
( ) |
( ) |
( ) |
|
Are all the cited references relevant to the research? |
(x) |
( ) |
( ) |
( ) |
|
Is the research design appropriate? |
(x) |
( ) |
( ) |
( ) |
|
Are the methods adequately described? |
(x) |
( ) |
( ) |
( ) |
|
Are the results clearly presented? |
(x) |
( ) |
( ) |
( ) |
|
Are the conclusions supported by the results? |
(x) |
( ) |
( ) |
( ) |
Comments and Suggestions for Authors
-None

Reviewer 3 Report
It is to be welcomed that the authors have made some of the requested changes. However, there are some considerations below:
Why have the authors not considered describing the results in table 2 graphically?
What reference(s) have they used for the correlation thresholds?
Indicate in Table 1 with symbology the significant results.
In the results they still do not write down the magnitude of the correlations.
Translated with www.DeepL.com/Translator (free version)
Author Response
We thank the Journal of Functional Morphology and Kinesiology for considering this manuscript “Manuscript ID: jfmk-2147743“ for 2nd revision. We also thank both the editor(s) committee and reviewers for their time, efforts, and feedback they provided reviewing this manuscript. We found that the comments and feedback were valuable, and helpful, and improved the quality of the manuscript. All the comments were addressed, and changes were made in the manuscript file and highlighted in GREEN. The manuscript underwent proofing and copyediting by a professional English speaker.
Please be aware that there are changes to the manuscript as a response to reviewers’ comments and feedback.
The reviewers’ comments were addressed, and the author’s responses are below:
Comments and Suggestions for Authors
It is to be welcomed that the authors have made some of the requested changes. However, there are some considerations below:
Why have the authors not considered describing the results in table 2 graphically?
All the significant corrections were presented graphically in the result section.
What reference(s) have they used for the correlation thresholds?
Reference numbers 24 and 25:
Cohen , J, Cohen J, Cohen , J, Cohen J. Statistical Power Analysis for the Behavioral Sciences. 2nd ed., editor. Statistical Power Analysis for the Behavioral Sciences. Routledge. https://doi.org/10.4324/9780203771587: L. Erlbaum Associates; 1988.
Gignac GE, Szodorai ET. Effect size guidelines for individual differences researchers. Pers Individ Dif [Internet]. 2016;102:74–8. Available from: https://www.sciencedirect.com/science/article/pii/S0191886916308194
Indicate in Table 1 with symbology the significant results.
The asterisk (*) was added to tables 1 and 2 to indicate significant results.
In the results they still do not write down the magnitude of the correlations.
The correlations’ magnitudes have been added to the result section.

Reviewer 4 Report
The authors addressed all the comments and improved the manuscript.